# SMILE: Audio-visual Speech Recognition with Siamese Masked Interaction Learning

## Abstract

Audio-Visual Speech Recognition (AVSR) aims to improve the performance of Automatic Speech Recognition (ASR) by incorporating visual cues in addition to audio information. Synchronized visual information plays a pivotal role in differentiating phonetically similar words, and it is highly beneficial in scenarios with high levels of noise. In AVSR, the crucial aspect is establishing temporal correspondence while aligning the mutually complementary nature of audio and visual modalities. To this end, we propose the Siamese Masked Interaction LEarning (SMILE) framework, which integrates multimodal mask learning with the Siamese architecture. SMILE facilitates global interactions among audio-visual features and enables single-modal and cross-modal local alignment. In addition, we propose an adaptive dynamic multimodal fusion strategy that effectively captures the complementary relationship between the audio and visual modalities. With extensive experiments, our model SMILE, when tested with different model scales and noise levels, achieves state-of-the-art performance on LRS2 and LRS3 datasets under both low-resource and high-resource settings.

## 1 Introduction

Audio-Visual Speech Recognition (AVSR) (Shi et al., 2022a; Radford et al., 2022; Anwar et al., 2023) is an active and evolving research area derived from the Automatic Speech Recognition (ASR) task. It is primarily designed to address real-world scenarios with challenges such as high levels of noise and speech occlusion. It integrates synchronized visual information with audio features to enhance the recognition accuracy and robustness. Zhang et al. (2023) propose the hypothesis that the lip movements of speakers in videos carry valuable information pertaining to appearance and temporal dynamics, which has a closer relationship with speech features than facial or body gestures. Lip movements can mainly provide phonetic articulatory details, including the place of articulation for sounds like bilabial, labiodental, and dental. It also conveys emphasis or stress, providing visual cues for heightened articulation, intensity, or duration. Due to this, we use the extracted frame-wise lip-region features in our experiments. Lip movements primarily contribute to the improvement of Automatic Speech Recognition (ASR) in two main scenarios. Firstly, they offer distinct visual cues that aid in differentiating between speech words that sound similar. Secondly, they provide valuable visual cues that enhance speech recognition accuracy in the presence of various types of noise commonly encountered in real-world environments.

Effectively leveraging visual information and promoting interaction between the audio and video modalities remains an area that requires further exploration. Most AVSR methods (Shi et al., 2022a; Zhu et al., 2022) utilize early fusion by concatenating audio and video modalities. This can promote models to learn the global modality-general features but lacks the ability to align local modality-specific features. Other methods (Hu et al., 2023; Dai et al., 2023) utilize cross-modal attention and contrastive learning to enable interactive communication between audio and video streams. This can facilitate better alignment of local modality-specific features but may suffer from insufficient global interactions. According to Li et al. (2022), the human brain will initially gather audio and visual information together and repeatedly interact between the two modalities, indicating the importance of early fusion and deep interaction. Inspired by this, we pro-

pose a Siamese Masked Interaction LEarning (SMILE)[1] framework to implement the multimodal interaction in an end-to-end manner, allowing for effective interactions throughout the process.

Siamese network (Bromley et al., 1993; Chopra et al., 2005; Koch et al., 2015; Chen & He, 2020; Grill et al., 2020; Caron et al., 2021; Zbontar et al., 2021) is a class of network architectures containing two or more identical sub-networks. It is a contrastive-based self-supervised learning method and can be viewed as an augmentation technique to enhance the semantic alignment capability of a model. In recent years, several methods have combined mask reconstruction with the Siamese architecture (Mishra et al., 2022; Assran et al., 2022; Shi et al., 2022c; Tao et al., 2022). These methods make the model establish stronger spatial or temporal contextual correlation while enhancing semantic alignment capability. In contrast, the purpose of the proposed SMILE framework is to enable single-modal interaction, cross-modal interaction, and global modal interaction simultaneously. This is achieved by carefully designed transformer attention layers and mask learning mechanisms. Fig. 1 depicts the three types of modality interactions, which will be demonstrated in detail in section 3.2.

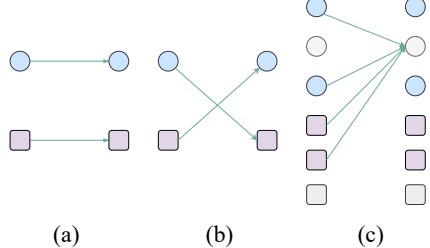

Figure 1: **Modal interaction types**. (a) Single-modal interaction. (b) Cross-modal interaction. (c) Global modal interaction. Gray shapes represent masked modalities. The masked output modality interacts with all of its unmasked input modalities.

## 2 RELATED WORKS

### 2.1 AUDIO-VISUAL SPEECH RECOGNITION

Most existing approaches try to make a better fusion between audio and video modalities with different model architectures and training strategies. Several methods have proposed utilizing transformer self-attention and cross-attention mechanisms between the audio and video modalities to enhance their interaction (Hu et al., 2023; Dai et al., 2023). Alternative approaches introduced prompts (Lin et al., 2023) or adapters (Thomas et al., 2022) to enhance the fusion of audio-visual multimodal information. Cheng et al. (2023a) and Cheng et al. (2023b) employed distinct training strategies to further align visual features in the multimodal feature space. To train a large multimodal model from scratch, Shi et al. (2022a) employed a clustering-based unsupervised pretraining approach and Shi et al. (2022b) further explored the noise robustness of audio-visual multimodal models. Haliassos et al. (2023) applied both intra-modality and inter-modality distillation techniques for multi-task learning. Zhu et al. (2022) designed a three-pathway multimodal model for three modalities. Obtaining aligned and labeled data in multimodality is rather cost-demanding.

### 2.2 SIAMESE MASKED INTERACTION LEARNING

The original Siamese networks (Bromley et al., 1993; Chopra et al., 2005; Koch et al., 2015; Chen & He, 2020; Grill et al., 2020; Caron et al., 2021; Zbontar et al., 2021) use two identical sub-networks for multi-view data augmentation, combined with stop-gradient operation, Exponential Moving Average (EMA) momentum update or other techniques to prevent feature collapse. In recent years, Masked Image Modeling (MIM) has emerged as a highly efficient method for semantic modeling (He et al., 2021; Huang et al., 2023), yielding remarkable results across various tasks. Several approaches (Mishra et al., 2022; Assran et al., 2022; Shi et al., 2022c; Tao et al., 2022; Wu et al., 2023) have been proposed to effectively integrate Siamese networks and masked image modeling, leading to promising results. These approaches attempted to mask image patches either in both views of the Siamese network (Mishra et al., 2022) or in a single view (Assran et al., 2022; Tao

---

[1] The term "interaction learning" specifically refers to the learning process of capturing the interplay between different modalities, distinguishing it from the conventional notion of learning through human interactions, commonly known as interactive learning.

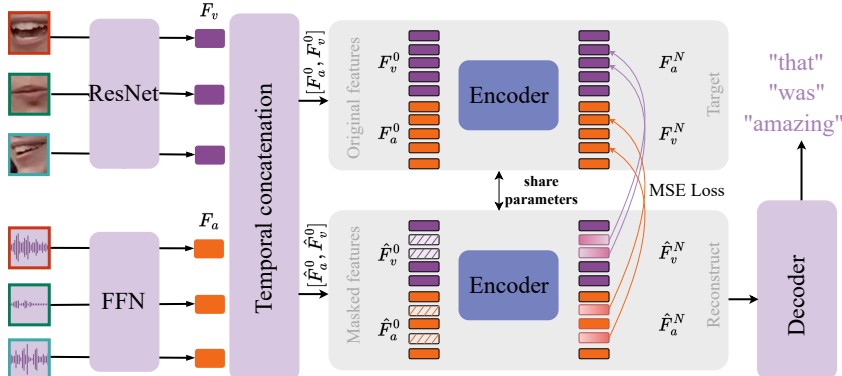

Figure 2: **The overall model pipeline**. Top branch: the target encoder. Bottom branch: the online encoder. The target encoder and online encoder share the same set of parameters. Our model benefits from early fusion and alignment interactions, achieving better audio-visual multimodal fusion.

et al., 2022), employing an aggressive masking rate (Wu et al., 2023). In our work, we use Siamese masked interaction learning in audio-visual speech recognition.

## 3 METHOD

### 3.1 PRELIMINARIES

The task of AVSR is to use audio-visual utterance pairs as input and generate the corresponding text as output. We extract audio and video features after data processing. As shown in Fig. 2, we use one linear projection layer for the audio front-end followed by layer normalization (Ba et al., 2016). And the generated speech feature is denoted by $F_a = \{F_a^t\}_{t=0}^{T-1} \in R^{T \times B \times C}$, where $T$ represents the temporal length, $B$ represents the batch size and $C$ represents the channel numbers. For video front-end, $F_v = \{F_v^t\}_{t=0}^{T-1} \in R^{T \times B \times C}$ is extracted by the modified ResNet-18 (Shi et al., 2022a) containing both 2D and 3D convolutions. Note that the sampling rate of speech is higher than that of video, so we stack several extracted audio frames in the temporal dimension to align with one video frame. The target of the model is to generate the corresponding text $g = \{g_i\}_{i=0}^{G-1} \in R^G$ where $g_i$ represents the $i$-th word in the text with a total length of $G$.

### 3.2 SIAMESE MASKED INTERACTION LEARNING

The overall pipeline is depicted in Fig. 2. After feature extraction, we concatenate $F_a$ and $F_v$ along the temporal dimension for deep interactions. We first pass the concatenated features through the Siamese encoder architecture. The Siamese encoder consists of two branches, each taking different inputs while sharing parameters. We define the network that receives features without random masking as the target encoder and the network that processes features with random masking as the online encoder. We apply random masking on the concatenated features for the online encoder and get features $[\hat{F}_a, \hat{F}_v]$ for the two modalities. As for the target encoder, we pass the original $[F_a, F_v]$ through it to provide target representation labels for the reconstructed output of the online encoder. We apply stop gradient operation on the target encoder to prevent the trivial constant embedding mentioned in Chen & He (2020). Let $[F_a^0 \leftarrow F_a, F_v^0 \leftarrow F_v]$ be the input for the target encoder and $[F_a^N, F_v^N]$ be the output for the target encoder, where $\leftarrow$ indicates the tensor copy operation and $N$ denotes the number of encoder layers. Similarly, $[\hat{F}_a^0 \leftarrow \hat{F}_a, \hat{F}_v^0 \leftarrow \hat{F}_v]$ and $[\hat{F}_a^N, \hat{F}_v^N]$ are the input and output of the online encoder. Assuming that $idx_a$ and $idx_v$ represent the indexes of masked tokens in the audio and video feature sequences, the Masked Representation Modeling (MRM) object can be described as follows:

$$L_{mrm} = \text{MSE}(F_a^N[idx_a], \hat{F}_a^N[idx_a]) + \text{MSE}(F_v^N[idx_v], \hat{F}_v^N[idx_v]). \tag{1}$$

We use Mean Square Error (MSE) to calculate the loss between masked tokens and their target labels. Note that the target labels are generated through the global multimodal interaction process,

where each token has already obtained rich semantic information from a global multimodal perspective. Thus, the Siamese architecture with masked representation modeling can make the model perform global modal interactions by learning to reconstruct the masked tokens in the corresponding positions (see Fig. 1(c)). Besides, the Siamese architecture enables learning a similarity metric by comparing pairs of corresponding representation views derived from the same unimodal feature. This enables the network to generalize effectively, even when trained with limited data.

Though mask learning enforces global modal interaction, reconstructing masked tokens from existing modality tokens is highly challenging. It requires a heavy interaction between the two modalities for modality-general and modality-specific information extraction. To strengthen the ability of single-modal and cross-modal interaction and alignment, we propose the mix-attention encoder layer. We apply a random masking strategy for each modality along the temporal dimension with different masking rates, where different masking lengths are applied at each randomly selected masking position. We define $M$ as the masked tokens. The feature representation for each modality with random masking can be expressed as follows:

$$\hat{F}_a = [F_{a_1}, F_{a_2}, \mathcal{M}, ..., F_{a_{t-2}}, \mathcal{M}, F_{a_t}], \hat{F}_v = [F_{v_1}, F_{v_2}, F_{v_3}, ..., \mathcal{M}, F_{v_{t-1}}, F_t]. \tag{2}$$

Let $(Q_a, K_a, V_a)$ denote the (query, key, value) triplet attention projection of audio features, and $(Q_v, K_v, V_v)$ denote the triplet for video features. We define $(Q_z, K_z, V_z)$ as follows:

$$Q_z = [Q_a^m, Q_a^u, Q_v^m, Q_v^u], K_z = [K_a^m, K_a^u, K_v^m, K_v^u], V_z = [V_a^m, V_a^u, V_v^m, V_v^u]. \tag{3}$$

In the equations above, every $Q_z$ is the concatenation (denoted by [...]) of audio and video features containing both masked feature tokens $Q_a^m, Q_v^m$ and unmasked feature tokens $Q_a^u, Q_v^u$. And the same is for $K_z$ and $V_z$. Note that the relative position between each masked and unmasked token is random. For clarity of explanation, the $Q_a^m$ and $Q_a^u$ in equation 3 only indicate the two types of tokens included in $Q_a$, without implying any specific order. The output of the attention operation $A_z$ can be obtained via:

$$A_z = \text{Softmax}(\frac{Q_z K_z^T}{\sqrt{D}})V_z, \tag{4}$$

in which $D$ represents the normalization factor. To further explain the single-modal and cross-modal interactions between audio and video masked and unmasked features, we conduct a more in-depth analysis below. Define $R_z$ as the correlation map calculated in the attention calculation, it can be described as:

$$
\begin{aligned}
R_z &= \text{Softmax}\left(\frac{Q_z K_z^T}{\sqrt{D}}\right) \\
&= \text{Softmax}\left(\frac{\begin{bmatrix} Q_a^m K_a^m & Q_a^m K_a^u & Q_a^m K_v^m & Q_a^m K_v^u \\ Q_a^u K_a^m & Q_a^u K_a^u & Q_a^u K_v^m & Q_a^u K_v^u \\ Q_v^m K_a^m & Q_v^m K_a^u & Q_v^m K_v^m & Q_v^m K_v^u \\ Q_v^u K_a^m & Q_v^u K_a^u & Q_v^u K_v^m & Q_v^u K_v^u \end{bmatrix}}{\sqrt{D}}\right) \\
&\triangleq \begin{bmatrix} R_{aa}^{mm} & R_{aa}^{mu} & R_{av}^{mm} & R_{av}^{mu} \\ R_{aa}^{um} & R_{aa}^{uu} & R_{av}^{um} & R_{av}^{uu} \\ R_{va}^{mm} & R_{va}^{mu} & R_{vv}^{mm} & R_{vv}^{mu} \\ R_{va}^{um} & R_{va}^{uu} & R_{vv}^{um} & R_{vv}^{uu} \end{bmatrix}.
\end{aligned} \tag{5}
$$

We have omitted the transpose operation $T$ for simplicity. From the audio perspective, $R_{aa}^{xy}$ denotes the single-modal similarity, and $R_{av}^{xy}$ denotes the cross-modality similarity. $R_{aa}^{mm}$ measures similarity between any masked tokens and $R_{aa}^{um}$ measures the similarity between unmasked and masked tokens. The same observations also apply to the relation map of video tokens. It shows that each modality simultaneously conducts self-attention and cross-modal attention, promoting single-modal and cross-modal interactions. As shown in Fig. 3, we then send the calculated $A_z^i$ for the $i-th$ layer into the Feed-Forward Network (FFN) to obtain the output $[\hat{F}_a^{i+1}, \hat{F}_v^{i+1}]$, which serves as input for the next mix-attention encoder layer.

The decoder component is a transformer auto-regressive decoding structure. We set the number of decoder layers to be half the number of encoder layers. The end-to-end encoder-decoder model is supervised by the cross-entropy loss denoted as $L_{ce}$, which is computed between the model's output and the corresponding text labels. This loss function measures the dissimilarity between

the predicted output and the ground truth labels, guiding the model to minimize the discrepancy and optimize its performance. Overall, the final loss $L$ is composed of supervised loss $L_{ce}$ and self-supervised loss $L_{mrm}$, which is defined as:

$$L = \alpha L_{mrm} + \beta L_{ce}, \tag{6}$$

in which $\alpha$ and $\beta$ are weight parameters to balance the contribution of the two losses. We introduce both self-supervised and supervised learning in our model, which can improve the model's performance with better audio-visual multimodal interactions.

### 3.3 ADAPTIVE MULTIMODAL FUSION

Since representations with masked tokens can further enhance the generalization ability of the decoder, we take the representation output from the online encoder and send it to the decoder. Moreover, we apply adaptive multimodal fusion on the concatenated multimodal output to fuse them across modalities and different encoder layers. The output representation from every online encoder transformer layer is of shape $2T \times B \times C$, in which $2T$ represents the temporal combination of audio and video representations. In AVSR, video serves as a supplementary modality to complement and enhance the audio information. Due to this, we partition the concatenated representation into the audio and video representation component both with shape $T \times B \times C$, and introduce a dynamic scaling factor to add them together. This can adjust the contribution of video modality to the fusion process. The dynamic scaling factor $S$ is shared across all layers. The formulation for dynamic scaling and addition for every $i$-th encoder layer and the weighted sum process is described as follows:

$$\hat{F}^i_{av} = \hat{F}^i_a + S * \hat{F}^i_v,$$
$$\hat{F}_{fuse} = \sum_{i=1}^{N} w_i \hat{F}^i_{av}, \tag{7}$$

in which $\hat{F}^i_{av}$ represents the dynamic fusion results for audio and video modalities, and $N$ represents the total number of mix-attention encoder layers. As mentioned in the study by Yang et al. (2021), the shallower layers in audio models tend to prioritize speaker-related information, while the deeper layers tend to emphasize content-related information. Recently, in the image domain, Liu et al. (2023) also explored the application of multi-level feature fusion and demonstrated improved performance. Inspired by these works, we adopt a similar approach by applying a multi-layer weighted sum to aggregate all layers together, akin to the method proposed by Yang et al. (2021). The weight coefficient $w_i$ corresponds to layer $i$ in the weighted summation, determining the contribution of each layer's output $F^i_{av}$ to the final fused output $F_{fuse}$. Both the weighting factor $w_i$ and the scaling factor $S$ are set as learnable parameters with an initial value of 1. Throughout the training, the model can adaptively adjust these factors to learn the optimal combination that effectively utilizes the complementary information from the audio and visual modalities.

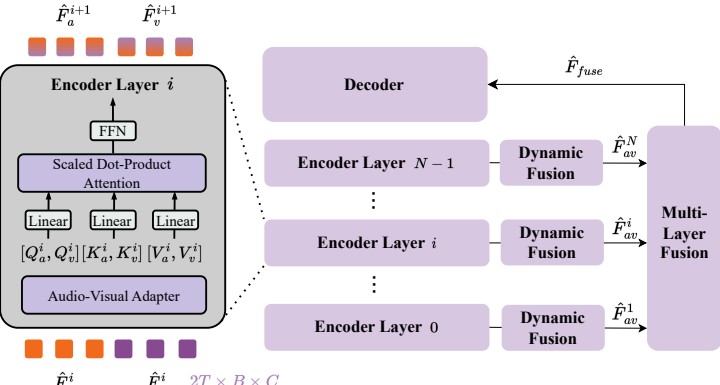

Figure 3: **Encoder layer and adaptive multimodal fusion.** Each encoder layer contains an additional audio-visual adapter to adapt the pre-trained weights. The dynamic fusion strategy is applied to the output $[\hat{F}^{i+1}_a, \hat{F}^{i+1}_v]$ for each mix-attention encoder layer $i$.

The structure of adaptive multimodal fusion is shown in Fig. 3. Note that the Siamese masked interaction learning process uses the concatenated output representations $[\hat{F}_a^N, \hat{F}_v^N]$ without using any fusion strategy. As the reconstruction of masked tokens is extremely difficult, we try to initialize our model with the parameters from Shi et al. (2022a) for easier startup. The difference is that we concatenate audio and video features in the temporal dimension and get shape $2T \times B \times C$, while Shi et al. (2022a) concatenate the two features in the channel dimension and get shape $T \times B \times 2C$. The difference in concatenation may produce significant biases during the attention operation in the transformer layer. Thus, we add the audio-visual adapter at the front of the transformer encoder layer. It has the same structure as the one proposed by Houlsby et al. (2019), which can adapt to changes in concatenation dimensions.

## 4 Experiments

### 4.1 Evaluation Metrics

In this paper, we measure the performance of Siamese masked interaction learning on AVSR tasks with two datasets. The evaluation metric used in speech recognition is the Word Error Rate (WER), which is defined as WER $= (P + D + I)/G$, where $P, D, I$ denote the counts of words replaced, deleted, and inserted, respectively, and $G$ represents the total number of reference words.

### 4.2 Implementation Details

We take Shi et al. (2022a) for AVSR with temporal-wise feature concatenation and additional audio-visual adapters as our baseline. Since lip movement exhibits the highest correlation with speech among visual cues, we only extract the lip region as visual speech input for our model. Moreover, we first detect 68 facial key points for the image face using dlib (King, 2009) and then align the profile face frames found in the video with its neighboring frontal faces. Next, we crop a $96 \times 96$ Region-Of-Interest (ROI) in the lip region. Also, we randomly crop a region of $88 \times 88$ from the entire ROI and perform a horizontal flip with probability 0.5 for data enhancement. As for the audio part, we extract a 26-dimensional log filter-bank energy feature at a stride of 10 ms with a sample rate of 16,000 Hz from the input raw waveform. Since image frames are sampled at 25Hz, we stack four adjacent acoustic frames together to synchronize the two modalities.

### 4.3 Experimental Results

We show our results on the LRS2 (Afouras et al., 2018a) and LRS3 (Afouras et al., 2018b) datasets (more information about the datasets can be found in Appendix D). We train our model under different model scales and dataset sizes with the parameters initialized in Shi et al. (2022a). The model scales include transformer-base and transformer-large. The transformer-base with blocks/embedding dimension/feed-forward dimension/attention heads in each transformer block is 12/768/3072/12. The transformer-large with blocks/embedding dimension/feed-forward dimension/attention heads in each transformer block is 24/1024/4096/16. Regarding the dataset size, we consider both low-resource and high-resource scenarios. The low-resource setting involves using 29 hours of labeled data from LRS2 and 30 hours of labeled data from LRS3, while the high-resource setting utilizes larger 224 hours of labeled data from LRS2 and the full 433 hours of labeled data from LRS3. Table 1 presents the results of our experiments, showcasing the performance under different combinations of model scales and dataset sizes.

As shown in Table 1, we compare our model with the previous methods on LRS2 and LRS3 datasets for the task of AVSR. It can be observed that our method can achieve state-of-the-art performance under different model scales and dataset sizes. This shows that our model can provide a better multimodal interaction strategy for audio and video modality with global and local single-modal and cross-modal interactions. Especially, our method demonstrates substantial advancements in scenarios with limited resources, indicating that the proposed Siamese architecture combined with masked representation modeling can effectively improve the model's generalization capability. The performance for the LRS2 low-resource setting is improved by 0.9% and 1.1% with transformer-base and transformer-large models, respectively.

Table 1: Experimental results on LRS2 and LRS3 clean datasets. LM denotes whether or not the model utilizes the language model. "Unlab hrs" denotes the number of unlabelled data hours used for pretraining. And "Lab hrs" denotes the number of labeled data hours used for finetuning. [*] denotes a seq-to-seq loss which is different from Cross-Entropy loss (CE). [†] is a technique where a model labels unannotated data using its own predictions to improve performance iteratively, which has proved effective for pretraining and usually improves the performance of downstream AVSR. We compare our method with both supervised and self-supervised methods. Besides, we show the model's performance under various settings of different model scales and data resource sizes.

| Method | Encoder | Criterion | LM | Unlab hrs | Lab hrs | WER(%) LRS2 | WER(%) LRS3 |
|---|---|---|---|---|---|---|---|
| *supervised* | | | | | | | |
| Afouras et al. (2018a) | Transformer | S2S[*] | ✓ | - | 224 | 8.5 | 8.3 |
| Afouras et al. (2018a) | Transformer | C2C | ✓ | - | 224 | 8.2 | - |
| Xu et al. (2020) | RNN | CE | ✗ | - | 590 | - | 7.2 |
| Petridis et al. (2018) | LSTM | CTC | ✓ | - | 380 | 7.0 | - |
| Yu et al. (2020) | TDNN | CTC | ✓ | - | 224 | 5.9 | - |
| Makino et al. (2019) | RNN | Transducer | ✓ | - | 31,000 | - | 4.8 |
| Ma et al. (2021) | Conformer | CTC+CE | ✓ | - | 380 | 4.7 | 3.2 |
| Hong et al. (2022) | Transformer | CE | ✗ | - | 224/433 | 4.5 | 3.4 |
| Hong et al. (2023) | Transformer | CE | ✗ | - | 224/433 | 4.1 | 2.8 |
| Hu et al. (2023) | Transformer | CE | ✗ | - | 224 | 3.1 | - |
| Serdyuk et al. (2021) | Transformer | - | ✗ | - | 90,000 | - | 2.3 |
| *self-supervised* | | | | | | | |
| **Base models, less training data** | | | | | | | |
| Hsu et al. (2021) | Transformer-Base | CE | ✗ | 1,759 | 30 | - | 5.0 |
| Zhu et al. (2022) | Transformer-Base | CE | ✗ | 1759 | 30 | - | 4.0 |
| Shi et al. (2022a) | Transformer-Base | CE | ✗ | 1,759 | 29/30 | 5.8 | 3.8 |
| Haliassos et al. (2023) | Transformer-Base | CTC+CE | ✗ | 1759 | 30 | - | 3.8 |
| SMILE (Ours) | Transformer-Base | CE | ✗ | 1,759 | 29/30 | **4.9** | **3.5** |
| **Large models, less training data** | | | | | | | |
| Hsu et al. (2021) | Transformer-Large | CE | ✗ | 1759 | 30 | - | 3.2 |
| Shi et al. (2022a) | Transformer-Large | CE | ✗ | 1759 | 29/30 | 4.8 | 2.9 |
| Zhu et al. (2022) | Transformer-Large | CE | ✗ | 1759 | 30 | - | 2.7 |
| Haliassos et al. (2023) | Transformer-Large | CTC+CE | ✗ | 1759 | 30 | - | 2.7 |
| SMILE (Ours) | Transformer-Large | CE | ✗ | 1759 | 29 | **3.7** | **2.5** |
| **Large models, more training data** | | | | | | | |
| Shi et al. (2022a) | Transformer-Large | CE | ✗ | 1,759 | 224/433 | 2.5 | 1.3 |
| Hsu & Shi (2022) | Transformer-Large | CE | ✗ | 1,759 | 433 | - | 1.2 |
| Cheng et al. (2023a) | Transformer-Large | CE | ✗ | 1,759 | 224 | 2.7 | - |
| Pan et al. (2022) | Transformer | CE | ✗ | 60000 | 224 | 2.6 | - |
| Zhu et al. (2022) | Transformer-Large | CE | ✗ | 1,759 | 224/433 | 2.3 | 1.2 |
| Haliassos et al. (2023) | Transformer-Large | CTC+CE | ✗ | 1,759 | 224/433 | 2.5 | 1.4 |
| Haliassos et al. (2023) w/ self-training[†] | Transformer-Large | CTC+CE | ✓ | 1,759 | 224/433 | 2.3 | 1.4 |
| Zhu et al. (2022) | Transformer-Large | CE | ✗ | 1759 | 224/433 | 2.3 | 1.2 |
| SMILE (Ours) | Transformer-Large | CE | ✗ | 1,759 | 224/433 | **2.2** | 1.2 |

Table 2: Experimental results on LRS2 dataset with musan noises. The number of mixture noise is set to 1. The noise probability is set to 1, which means that all test speeches are added with noise. Note that both (Hsu et al., 2021) and our model are trained under same noise setting.

| Model | Encoder | Lab hrs | Test noise snr -10 | -5 | 0 | 5 | 10 | WER Avg. |
|---|---|---|---|---|---|---|---|---|
| *Base models, less training data* | | | | | | | | |
| Hsu et al. (2021) | Transformer-Base | 29h | 34.69 | 19.06 | 11.65 | 8.51 | 6.98 | 16.18 |
| Smile(Ours) | Transformer-Base | 29h | 24.36 | 14.32 | 9.02 | 7.47 | 6.42 | 12.32 |
| *Large models, less training data* | | | | | | | | |
| Hsu et al. (2021) | Transformer-Large | 29h | 22.65 | 13.82 | 8.28 | 7.71 | 7.03 | 11.90 |
| Smile(Ours) | Transformer-Large | 29h | 20 | 10.76 | 6.69 | 5.67 | 5.3 | 9.68 |
| *Large models, more training data* | | | | | | | | |
| Hsu et al. (2021) | Transformer-Large | 224h | 12.22 | 7.34 | 4.39 | 3.24 | 3.07 | 6.05 |
| Smile(Ours) | Transformer-Large | 224h | 11.05 | 6.68 | 4.12 | 3.14 | 3.02 | 5.60 |

We suppose that lip movments can provide valuable visual cues that enhance speech recognition accuracy in various real-world noises. To validate the performance of our model under noise conditions, we use musan dataset (Snyder et al., 2015) combined with LRS3 (Afouras et al., 2018b) speech noise as our noise dataset. We trained our model on LRS2 dataset with different model settings. For each speech segment in the dataset, we intentionally introduced a random type of noise from musan, setting the signal-to-noise ratio (SNR) to 0. The experimental results under noise scenarios are shown in 2.

## 4.4 ABLATION STUDY

To validate the effectiveness of our model, we conduct a comparative analysis between our default setting and other alternative approaches.

**Effectiveness of different components.** As shown in Table 3, we compare different parts with the baseline. We take Shi et al. (2022a) for AVSR with temporal-wise feature concatenation and additional audio-visual adapters as our baseline. The model's performance improves after incorporating adaptive multimodal fusion, indicating an enhanced ability to capture deep complementary relationships between the audio and video modalities. Then, we present results when masking is applied solely to the features as a form of feature augmentation rather than incorporating it with the Siamese architecture. The observed improvement resulting from the masked feature augmentation provides evidence of its effectiveness in enhancing the model's ability to capture temporal dependencies on features by mitigating temporal feature redundancy. Furthermore, we combine the masked data augmentation with Siamese architecture. We find that the Siamese masked interaction learning strategy improves the model's performance by a large margin of 0.7% ($4.4\% \rightarrow 3.7\%$) and 0.3% ($2.8\% \rightarrow 2.5\%$) separately, which shows its effectiveness of learning audio-visual multimodal interactions.

Table 3: Ablation study for the efficacy of different components in our method. We use transformer-large model for LRS2 and LRS3 low-resource datasets.

| Method | WER(%) | |
|---|---|---|
| | LRS2 | LRS3 |
| Shi et al. (2022a) + Audio-visual adapter (Baseline) | 4.8 | 3.0 |
| + Adaptive multimodal fusion | 4.6 | 2.9 |
| + Masked feature augmentation | 4.4 | 2.8 |
| + Siamese masked interaction learning | 3.7 | 2.5 |

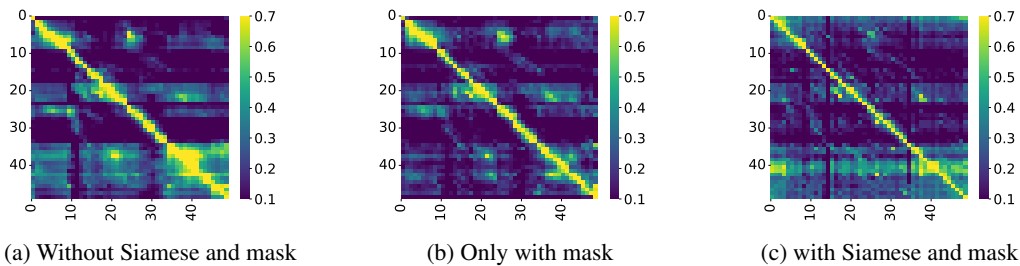

(a) Without Siamese and mask      (b) Only with mask      (c) with Siamese and mask

Figure 4: The efficacy of Siamese masked interaction learning for audio-visual temporal correlation. Each row in the visualization corresponds to the temporal token index of the video representations, while each column represents the temporal token index of the audio representations.

To further demonstrate the effectiveness of Siamese masked interaction learning, we visualize the cosine correlation between the audio and video representations at the output of the encoder transformer layer. As depicted in Fig. 4, we present the results for the model trained under three scenarios: without Siamese and mask, only with mask, and with Siamese and mask. The diagonal of the visualization represents the corresponding pairs of audio and video tokens, while the other parts represent the pairs of audio and video tokens without temporal correspondence. The visualization

Table 4: Ablation study on Siamese training strategies and masking rate ablation. The ablations for Siamese training strategies are under the low-resource LRS2 29h dataset with transformer-base model. The ablations for masking rates and lengths are under low-resource LRS3 30h with transformer-base model.

| (a) Siamese training strategies. | | | | (b) Masking rate ablations. | | | | |
|---|---|---|---|---|---|---|---|---|
| **Siamese architecture** | | **WER(%)** | | **Masking rate** | | | | **WER(%)** |
| Target encoder | Online encoder | | | Speech | Length | Video | Length | |
| (1) Finetune | Finetune | 6.71 | | 0.3 | 6 | 0.3 | 6 | 3.71 |
| (2) Stop gradient | Finetune | 4.94 | | 0.4 | 12 | 0.6 | 6 | 3.69 |
| (3) EMA update | Finetune | 5.18 | | 0.6 | 12 | 0.4 | 6 | 3.54 |
| (4) Initially frozen | Finetune | 5.28 | | 0.8 | 12 | 0.6 | 6 | 3.72 |

clearly illustrates that the Siamese architecture combined with masked interaction learning can effectively promote the temporal alignment between audio and video modalities, capturing the deep correspondence between them.

**Different training strategies.** The proposed Siamese architectures (Mishra et al., 2022; Assran et al., 2022; Shi et al., 2022c; Tao et al., 2022) usually apply different training strategies to the target network like stop gradient operation and Exponential Moving Average (EMA). Our model defines the network that receives features without the masking strategy as the target encoder and the network that processes features with random masking as the online encoder. In Table 4a, we mainly explore different training strategies for the target encoder. We first train the online encoder and target encoder with shared parameters simultaneously. Subsequently, we experiment with incorporating the stop gradient operation for the target network, resulting in improved results (see row 2 in Table 4a). Next, we explore updating the target network using the EMA momentum update, leading to a slower update pace for the target network than the online network. Additionally, we examine the scenario where the target encoder remains frozen from the beginning and is not updated during the training process. Notably, updating the target encoder at a slower rate or even not updating it at all (rows 3 and 4 in Table 4a) does not yield results as promising as the stop gradient operation. This observation suggests that the target encoder, with shared parameters, may offer more accurate labels for the masked representation predictions generated by the online encoder.

We apply the masking strategy for audio representation and video representation with different masking rates and lengths at each masking position. As shown in Table 4b, the optimal performance of the model is achieved when 60% of the audio representations and 40% of the video representations are masked. We observe that masking a higher audio percentage than video yields better results. This can be attributed to the fact that capturing the temporal dependencies in audio by reconstructing the masked tokens with audio-visual interactions globally and locally is crucial for achieving optimal model performance. In addition to the masking rate for each modality, the results indicate that the performance remains satisfactory even with a masking length of up to 12 in the audio representation at each masking position. It suggests that there is significant temporal redundancy in audio features. Therefore, eliminating this redundancy and enhancing the model's temporal modeling capacity by reconstructing with multimodal interactions are necessary.

## 5 CONCLUSION

In this paper, we propose the Siamese Masked Interaction Learning (SMILE) framework for AVSR. SMILE facilitates global interactions among audio-visual features and enables single-modal and cross-modal local alignment. Experimental results on two public datasets LRS2 and LRS3 show that our method can achieve state-of-the-art performance under different dataset scales and different numbers of labeled data. Our model demonstrates significant performance improvements, specifically in resource-constrained settings such as LRS2 (29 hours) and LRS3 (30 hours). In future work, we will try other audio-visual multimodal tasks. Since the language model has demonstrated tremendous power recently, we will explore the pre-trained model combined with three modalities of audio, video and language in the future.

ETHICS STATEMENT

All data utilized in this study are publicly available and have been obtained under the following three licenses: TED terms of use, Creative Commons BY-NC-ND 4.0 license, and Creative Commons Attribution 4.0 International License. We have conducted spot-checking and found that the datasets exhibit gender balance and encompass a diverse range of races and ages. However, it should be noted that the distribution of speakers in the data may not be representative of the global human population, potentially introducing unintended biases related to societal, gender, racial, and other factors. To ensure anonymity, the visuals in the paper exclusively focus on the mouth area of speakers whenever they are depicted. It is important to exercise caution regarding unintentional biases that may arise from this fact. The proposed method has potential applications in various domains, including security and crime investigations. However, it is crucial to acknowledge the potential for misuse, such as surveillance and wiretapping. We are committed to responsible distribution of our code and model, taking special care to address any potential security and privacy concerns that may arise.

REPRODUCIBILITY

To ensure reproducibility, we provide as many implementation details as possible in the main paper as well as tables showing the hyperparameter values in the appendix. Moreover, we plan on making the code and pre-trained models publicly available.

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

## A ABLATION RESULTS

Table 5: Ablations for the Siamese masked interaction learning architecture and the total loss weight. * in ablation table (a) refers to padding the masked speech tokens with temporally aligned video tokens.

(a) Masking types.

| Padding types | WER(%) |
|---|---|
| Zero | 4.9 |
| Noise | 5.4 |
| Correspondence* | 5.2 |

(b) Masking strategies.

| Loss | WER(%) |
|---|---|
| Masked | 4.9 |
| All | 5.1 |

(c) Prediction heads for loss.

| Prediction heads | WER(%) |
|---|---|
| Both | 5.1 |
| Online Network | 5.0 |
| None | 4.9 |

(d) Loss weight parameters.

| Alpha | Beta | WER |
|---|---|---|
| 0.8 | 0.2 | 5.66 |
| 0.5 | 0.5 | 5.55 |
| 0.2 | 0.8 | 5.59 |

More ablations about Siamese masked interaction learning can be found in Table 5. We present ablation studies involving diverse masking types, masking strategies, and the incorporation of prediction heads for loss calculation. In Table 5a, we try different masking types and find that zero padding shows the best performance. We hypothesize that zero padding helps maintain consistency and smoothness in the multimodal training process. Table 5b, shows that the reconstruction loss with only masked indexed tokens performs better than loss with all tokens, which may lead to a trivial solution for the Siamese architecture. In Table 5c, we try to add prediction heads aside for loss calculation. And we find that no prediction heads for the online network or the teacher network perform best. In Table 5d, we investigated the impact of the weight relationship between two different losses on the experimental results.

## B AUDIO-VISUAL AND AUDIO-ONLY COMPARISON

Table 6: Audio-visual and audio-only comparison. For the audio-visual test and audio-only test settings, the models are trained using the noise-pro set to 1 and noise-snr set to 0. And test the model with audio-visual and audio-only conditions. For the Audio-only train&test setting, the model is both trained and tested with only the audio modality.

| Model | Settings | Clean WER | Test noise snr | | | | | Noise WER |
|---|---|---|---|---|---|---|---|---|
| | | | -10 | -5 | 0 | 5 | 10 | |
| Hsu et al. (2021) | Audio-visual test | 6.42 | 34.69 | 19.06 | 11.65 | 8.51 | 6.98 | 16.18 |
| | Audio-only test | 7.28 | 82.01 | 54.84 | 26.11 | 14.21 | 9.53 | 37.34 |
| | Audio-only train&test | 7.79 | 71.75 | 44.5 | 20.52 | 12.28 | 8.87 | 31.58 |
| Smile(Ours) | Audio-visual test | 5.55 | 24.36 | 14.32 | 9.02 | 7.47 | 6.42 | 12.32 |
| | Audio-only test | 11.02 | 79.56 | 53.52 | 30.78 | 19.45 | 14.42 | 39.55 |
| | Audio-only train&test | 7.02 | 72.62 | 43.84 | 20.24 | 12.13 | 8.75 | 31.52 |

## C  TRAINING SETTINGS

Table 7: Training settings.

| Hyperparameter | Tansformer-base value | Transfomer-large value |
|---|---|---|
| Max tokens | 1000 | 1000 |
| Max steps | 30k | 18k |
| Learning rate scheduler | tri stage | tri stage |
| Warmup steps | 10k | 6k |
| Decay steps | 20k | 12k |
| Learning rate encoder | 1e-4 | 1e-4 |
| Learning rate decoder | 1e-3 | 1e-3 |
| Optimizer ($\beta_1$,$\beta_2$) | (0.9,0.98) | (0.9,0.98) |
| Encoder blocks | 12 | 24 |
| Encoder hidden size | 768 | 1024 |
| Encoder FFN hidden size | 3072 | 4096 |
| Encoder attention heads | 12 | 16 |
| Decoder blocks | 6 | 12 |
| Decoder hidden size | 768 | 1024 |
| Decoder FFN hidden size | 372 | 4096 |
| Decoder attention heads | 4 | 8 |

## D  DATASETS

**LRS2-BBC** The Oxford-BBC Lip Reading Sentences 2 (Afouras et al., 2018a) dataset is a publicly available English lip-reading dataset that is commonly used for research in the field of lip-reading and audio-visual speech recognition. This dataset contains a large amount of video data extracted from BBC television shows, which cover a wide range of topics and have diverse speakers. The videos are transcribed at the sentence level, aligning every spoken word with the corresponding text. LRS2 consists of approximately 224 hours of training data, including 195 hours of pretraining data and 29 hours of trained data. One notable difference between the pretraining and training partitions is that the video clips in the pretraining partition are not strictly trimmed. This means the video duration in the pretraining partition may be longer than the corresponding sentence text.

**LRS3-TED** Lip Reading Sentence 3 dataset (Afouras et al., 2018b) has the same structure as the LRS2-BBC dataset. It includes 433 hours of video extracted from TED and TEDx talks, along with the corresponding subtitles and word alignment boundaries. Similarly, 433h of the pretraining data is for high-resource training and 29h for low-resource training.

## E  RESULTS COMPARISONS

As shown in Fig. 5, we choose some results and provide the corresponding lip-movement visualizations. Note that we only chose some short audio-visual pairs to have a clearer look at the variations in the lip shape and better understand how lip movements affect the accuracy of recognition results. The first two examples are cases where Siamese masked interaction learning can enhance the utilization of lip-reading information by the model. The following two examples demonstrate that combining the finetuned model with the Siamese masked interaction learning method can result in excessive reliance on lip information. We can observe that mismatched lip movements with normal pronunciation can lead to recognition errors. The last two examples indicate some challenging cases where correct recognition is not achieved even with the inclusion of the Siamese masked interaction learning method. However, we can still observe that integrating Siamese masked interaction learning provides a certain level of corrective effect. We can see that incorporating the Siamese masked interaction learning method allows the model to establish better connections between audio and video, resulting in improved multimodal fusion.

Figure 5: Transcriptions from different audio-visual speech recognition models. GT: ground-truth, Proposed: SMILE training, Supervised: original model with decoder. Red: wrong words in the output

| | | Correct with Siamese masked interaction learning |
|---|---|---|
| (1) | GT: | let's just calm down |
| | Supervised: | let's just come down |
| | Proposed: | let's just calm down |

| (2) | GT: | end of november |
|---|---|---|
| | Supervised: | in north november |
| | Proposed: | end of november |

| | | Wrong with Siamese masked interaction learning |
|---|---|---|
| (3) | GT: | but you have to go |
| | Supervised: | but you have to go |
| | Proposed: | that you have to go |

| (4) | GT: | harold in neighbours |
|---|---|---|
| | Supervised: | harold in neighbours |
| | Proposed: | harold and neighbours |

| | | Both wrongly recognized cases |
|---|---|---|
| (5) | GT: | fame and fortune |
| | Supervised: | famous (and) fortune |
| | Proposed: | famous a fortune |

| (6) | GT: | the shot of musket |
|---|---|---|
| | Supervised: | the shared of muscar |
| | Proposed: | the shirt of musket |

