# OpenReview forum: "SMILE: Audio-Visual Speech Recognition with Siamese Masked Interaction Learning"
_ICLR.cc/2024/Conference — Submitted to ICLR 2024_

### Official Review · Reviewer_reNT · 2023-10-25

**Soundness:** 2 fair
**Presentation:** 3 good
**Contribution:** 2 fair
**Rating:** 5
**Confidence:** 4

**Summary:**

A transformer-based model is trained to perform audiovisual speech recognition, where fused features are learned by masking audio/visual tokens and then learning to reconstruct the masked tokens in a Siamese architecture.  The fused multimodal features from different network layers are then dynamically fused using adaptive weighting, before being passed to the decoder network for recognition.

**Strengths:**

- The work is based on open-datasets, and authors propose to release their code and pre-trained models.  This all enables reproducibility.
- Although computing a weighted fusion of features from different network layers already has been shown to be effective in ASR, the utility is further demonstrated here using multimodal features rather than acoustic-only features.
- The writing quality is good, and the paper is well structured.

**Weaknesses:**

- It is not clear to me how much the visual information contributes to the WER.  The WERs reported here are on par with state of the art acoustic-only speech recognition.  Why not include an ablation of the modalities — show results for audio-only, video-only and audiovisual.
- Also, the motivation for the work is that the visual modality helps in the presence of acoustic noise.  There are no experiments that actually demonstrate this, or quantify the effective gain in SNR provided by the visual information.
- The paper describes the problem of AVSR as an “emerging research area”, but this ignores the 30+ years of prior research into multimodal speech recognition.

**Questions:**

I find the introductory paragraph that motivates multimodal speech recognition to be unclear.  The authors state that “... speech contains detailed phonetic information, while lip movement conveys emphasis or stress ...”.  Phonetic information is conveyed visually too — for example, the place of articulation for bilabial, labiodental, and dental sounds can be seen.  Furthermore, emphasis and stress are both also conveyed acoustically.  The main benefit of multimodal speech is that when acoustic noise masks the differences between two sounds so that they sound similar, those sounds can be disambiguated visually.  Similar sounding speech events look different, and similar looking speech events sound different from one another.  This is the complementary nature of audiovisual speech.

The authors refer to “The original Siamese networks” and cite papers from circa 2020, yet Siamese networks pre-dated these works by a couple of years.

Equation (7) and the accompanying explanation feels a little redundant as this already is clear from Equation (6).

In Equation (9), S is shared across all layers, which weighs the contribution of the visual information.  What about using a dynamic, layer-dependent weight (as in Equation (10)) since the contribution of the visual information for the speech recognition task might be different for different layers (as you already note for the acoustic information:  “... the shallower layers in audio models tend to prioritize speaker-related information, while the deeper layers tend to emphasize content-related information.”.  The same might be true for visual features, and allowing the visual contribution to \hat{F}^{I}_{av} to vary might provide more robust features.

In Section 4.2 you refer to extracting the per-frame ROI and then applying a horizontal flip with probability 0.5.  To be clear, this flip is not applied per frame, but to the whole sequence?

For Table 3(b), why were these four combinations of masking rate for the audio and visual selected?  What about the other combinations?

---

> ### Author Response · Authors · 2023-11-18
>
> Page [1/2]
>
> I'm gratitude for your valuable feedback and suggestions on our manuscript. We list our responses to the questions below.
>
> >Questions: I find the introductory paragraph that motivates multimodal speech recognition to be unclear. The authors state that “... speech contains detailed phonetic information, while lip movement conveys emphasis or stress ...”. Phonetic information is conveyed visually too — for example, the place of articulation for bilabial, labiodental, and dental sounds can be seen. Furthermore, emphasis and stress are both also conveyed acoustically. The main benefit of multimodal speech is that when acoustic noise masks the differences between two sounds so that they sound similar, those sounds can be disambiguated visually. Similar sounding speech events look different, and similar looking speech events sound different from one another. This is the complementary nature of audiovisual speech.
>
> ANS: I appreciate your clear and precise insights, which teach me a lot. According to your advise, we re-formulate our paper and provide   the corresponding experiments in our revised paper. Please see the first part of the introduction and new Table 2. Our motivation is : Lip movements primarily contribute to the improvement of Automatic Speech Recognition (ASR) in two main scenarios. Firstly, they offer distinct visual cues that aid in differentiating between speech words that sound similar. Secondly, they provide valuable visual cues that enhance speech recognition accuracy in the presence of various types of noise commonly encountered in real-world environments.
>
> >Questions: The authors refer to “The original Siamese networks” and cite papers from circa 2020, yet Siamese networks pre-dated these works by a couple of years.
>
> ANS: Thanks for pointing it out. We have corrected it in our revised paper.
>
> > Questions: Equation (7) and the accompanying explanation feels a little redundant as this already is clear from Equation (6).
>
> ANS: We believe that this formula better captures the interaction process between the audio and video modalities, specifically between masked tokens and unmasked tokens.
>
> >Questions: In Equation (9), S is shared across all layers, which weighs the contribution of the visual information. What about using a dynamic, layer-dependent weight (as in Equation (10)) since the contribution of the visual information for the speech recognition task might be different for different layers (as you already note for the acoustic information: “... the shallower layers in audio models tend to prioritize speaker-related information, while the deeper layers tend to emphasize content-related information.”. The same might be true for visual features, and allowing the visual contribution to \hat{F}^{I}_{av} to vary might provide more robust features.
>
> ANS: We agree with your viewpoint. However, when conducting experiments with specific S_i values for each layer, we observed that this specific S_i setting performed worse compared to sharing the same S value across all layers. We postulate that S may represent the global weight assigned to visual cues that need to be incorporated into speech.
>
> >Questions: In Section 4.2 you refer to extracting the per-frame ROI and then applying a horizontal flip with probability 0.5. To be clear, this flip is not applied per frame, but to the whole sequence?
>
> ANS: For each frame in the whole sequence, it has the chance to be flipped with a probability of 0.5.
>
> >Questions: For Table 3(b), why were these four combinations of masking rate for the audio and visual selected? What about the other combinations?
>
> ANS: We firstly apply 0.3 masking rate and 6 mask length for both audio and visual modalities as our base setting and find it works well. Then we suppose that Siamese architecture may work with more masking rate and masking length, which is benefit to improve the audio-visual interaction and learn a higher-level audio-visual temporal correspondence. Considering that audio feaures are more redundant than visual features in terms of temporal information, we raise audio mask length to 12. We compare audio-to-video mask ratio with 0.4:0.6 and 0.6:0.4 under same mask length setting, and find that 0.6:0.4 is better. We suppose that masking more audio can force the model to pay more attention to visual cues, which may reduce the “audio-predominant” modality-imbalance problem. Further increasing the masking rate may lead to significant loss of modality information, resulting in a decline in model performance.

---

> > ### Comment · Reviewer_reNT · 2023-11-23
> > **Comment by Reviewer reNT**
> >
> > Thanks for answering my questions and including the results from the additional experiments.
> >
> > Regarding flipping individual frames -- I am puzzled by flipping individual frames in the sequence with probability 0.5.  In the worst case every other frame is flipped, which in the context of lip motion for speech will result in just noise (given lip motion will not be symmetric about the vertical axis).  Have you watched any sequences back aligned with the audio to see what you're training your model with?

---

> > > ### Author Response · Authors · 2023-11-23
> > >
> > > Thank you for your reply.
> > >
> > > We apologize for any confusion caused. In the Siamese architecture, a combination of the online network and the target network is utilized. The online network incorporates the masking strategy and the horizontal-flip data augmentation technique, whereas the target network does not employ these strategies. We suppose that this approach can form a better data augmentation strategy and enhance the local modeling capability of the model.
> > >
> > > We appreciate your insightful analysis. We will implement the data augmentation strategy you have suggested and thoroughly evaluate its impact on the performance of the model.

---

> ### Author Response · Authors · 2023-11-18
>
> Page[2/2]
>
> Continue here.
> >Weaknesses: It is not clear to me how much the visual information contributes to the WER. The WERs reported here are on par with state of the art acoustic-only speech recognition. Why not include an ablation of the modalities — show results for audio-only, video-only and audiovisual.
>
> ANS: We provide Table 6 in the appendix in our newly revised paper.
>
> > Weaknesses: Also, the motivation for the work is that the visual modality helps in the presence of acoustic noise. There are no experiments that actually demonstrate this, or quantify the effective gain in SNR provided by the visual information.
>
> ANS: We newly provide Table 2 which is trained and tested on LRS2 with noise conditions.
>
> >Weaknesses: The paper describes the problem of AVSR as an “emerging research area”, but this ignores the 30+ years of prior research into multimodal speech recognition.
>
> ANS: Thanks for pointing it out. I revise it into "AVSR is an active and evolving research area derived from the Automatic Speech Recognition (ASR) task."

---

### Official Review · Reviewer_brXn · 2023-10-30

**Soundness:** 2 fair
**Presentation:** 3 good
**Contribution:** 3 good
**Rating:** 5
**Confidence:** 4

**Summary:**

The paper proposes methods to improve the performance of AVSR systems. The authors propose Simaese Masked Interaction Learning (SMILE) and adaptive multimodal fusion, which enables early fusion and representation alignment methods between the modalities through transformer layers and mask learning mechanisms. This facilitates single-modal interaction, cross-modal interaction and global modal interaction simultaenously. The authors perform AVSR experiments on LRS2 and LRS3 datasets on which they demonstrate performance exceeding existing work.

**Strengths:**

- The design choices are well-motivated, suitable for the problem of AVSR.
- The proposed model is trained on both low and high-resource settings, and show state-of-the-art performance on both.
- Ablations in Table 2 demonstrate that all of the design choices contribute positively to performance.

**Weaknesses:**

- Although the results are better than the previous works, the improvement is marginal and appears to be saturated.
- The authors state that AVSR "is primarily desgined to address ... high levels of noise and speech occlusion" but the paper does not address such scenarios.
- Related to above points, the LRS2/3 dataset was collected using an automated pipeline using ASR. This means that the test set is biased towards examples that are easy for speech recognition systems -- the speech-only model in the data curation pipeline can achieve 100% accuracy (Sec 4 of 1809.02108 and Sec 2.1 of 1809.00496). This undermines the usefulness of this dataset for AVSR under clean conditions.
- The authors cite 2020 and later papers for original Siamese network, but the term was proposed much before by (Koch et al., 2015) for contrastive learning.

**Questions:**

- Does the model work in missing modality conditions?
- What is the performance of audio-only ASR model trained using the same pretraining/finetuning data setup?
- How sensitive is the result to the values of alpha and beta.

---

> ### Author Response · Authors · 2023-11-18
>
> Thanks for your valuable advise, i appreciate it a lot. We list our responses to the questions below.
> >Weaknesses: The authors state that AVSR "is primarily desgined to address ... high levels of noise and speech occlusion" but the paper does not address such scenarios.
>
> ANS: We re-formulated our motivation in the revised paper to make it more clearer. Our motivation is revised in the first paragraph in the introduction part.
>
> >Weaknesses: Related to above points, the LRS2/3 dataset was collected using an automated pipeline using ASR. This means that the test set is biased towards examples that are easy for speech recognition systems -- the speech-only model in the data curation pipeline can achieve 100% accuracy (Sec 4 of 1809.02108 and Sec 2.1 of 1809.00496). This undermines the usefulness of this dataset for AVSR under clean conditions.
>
> ANS: We provide Table 2 in our paper, which is experimented on noise conditions.
>
> > Weaknesses: The authors cite 2020 and later papers for original Siamese network, but the term was proposed much before by (Koch et al., 2015) for contrastive learning.
>
> ANS: Thanks for pointing out it. We correct it in the revised paper.
>
> >Questions: Does the model work in missing modality conditions?What is the performance of audio-only ASR model trained using the same pretraining/finetuning data setup?
>
> ANS: In the newly-revised paper, Table 6 in the appendix is added to provide the audio-only test and audio-only train&test results of our model in both clean and noise test conditions. All models are trained under noise conditions, with a fixed number of mixtures set to 1, a probability of noise set to 1, and a signal-to-noise ratio (SNR) set to 0.
>
> >Questions: How sensitive is the result to the values of alpha and beta.
>
> ANS: In the newly-revised paper, Table 5(d) in the appendix is added.

---

> ### Comment · Reviewer_brXn · 2023-11-23
>
> Thanks for the revision. In the new Table 2, why is the SNR higher for the baseline (Hsu et al., 2021) compared to the proposed method? Isn't high SNR better?

---

> ### Author Response · Authors · 2023-11-23
>
> Thank you for your reply.
>
> In the new Table 2, the results are WER results under different SNR settings. For example, the test noise snr=-10 column represents that we add noise to the test speech with snr=-10 setting. Under this noise setting, we get the corresponding WER results. And the lower WER results means better model performance. The last column of Table 2 is the average WER result, which is the average WER for snr settings for -10,-5,0,5,10.

---

### Official Review · Reviewer_BvrV · 2023-11-01

**Soundness:** 3 good
**Presentation:** 3 good
**Contribution:** 3 good
**Rating:** 5
**Confidence:** 4

**Summary:**

This paper proposes a model named Siamese Masked Interaction LEarning
(SMILE), which combines the multimodal fusion and masked reconstruction for audio-visual speech recognition. SMILE model encodes both the original and masked features and is trained with a combination of cross-entropy loss and masked prediction loss. The model additionally combines the features from different transformer layers with a learnable weighted sum. The method is evaluated on LRS2 and LRS3 datasets and has outperformed several baselines in the two audio-visual speech recognition benchmarks.

**Strengths:**

The paper is overall well-written. The authors also conducted a thorough analysis to show the effectiveness of each component of the SMILE model.  The proposed method, including using masked reconstruction under supervised setting, is novel to audio-visual speech recognition, specifically. The ablation on training strategies of the Siamese network is also a potentially useful finding.

**Weaknesses:**

The proposed method falls behind state-of-the-art approaches. For example, [1] has achieved 0.9% WER in LRS3, which is better than the proposed model. In addition, the model is not evaluated under noisy setting, which is a typical use case of audio-visual speech recognition. In the broad context, the proposed method also lacks novelty. For example, the weighted combination of layerwise features is a widely used method in the SUPERB benchmark.

[1]. Pingchuan Ma, Alexandros Haliassos, Adriana Fernandez-Lopez, Honglie Chen, Stavros Petridis, and Maja Pantic. Auto-AVSR: Audio-visual speech recognition with automatic labels.

**Questions:**

See strengths and weakness.

---

> ### Author Response · Authors · 2023-11-16
>
> Thank you for giving us valuable comments. We present responses to the questions below.
>
> The method described in [1] used more training data than ours, as indicated by the data presented in Table 4 of [1]. A clearer   comparison is listed below.
> | Method | Data size |  Dataset  | WER |
> |---------|---------|---------|---------|
> |SMILE(Ours)|1759h|Voxceleb2,LRS3|1.2|
> |[1]|1902h|LRW,Voxceleb2,LRS3|1.0|
> |[1]|3448h|LRW,LRS2,Voxceleb2,LRS3|0.9|
>
> [1]. Pingchuan Ma, Alexandros Haliassos, Adriana Fernandez-Lopez, Honglie Chen, Stavros Petridis, and Maja Pantic. Auto-AVSR: Audio-visual speech recognition with automatic labels.

---

> > ### Comment · Reviewer_BvrV · 2023-11-21
> >
> > Thank you for addressing the comments. Under noisy setting, is there a comparison to audio-visual models (e.g., AV-HuBERT)?

---

> > > ### Author Response · Authors · 2023-11-22
> > >
> > > Thank you for your response. You can find this comparison in Table 2 of the revised manuscript. Additionally, all modified sections are highlighted in blue.

---

### Official Review · Reviewer_iBsX · 2023-11-06

**Soundness:** 3 good
**Presentation:** 3 good
**Contribution:** 2 fair
**Rating:** 5
**Confidence:** 4

**Summary:**

This paper tackles the problem of audio-visual speech recognition, and puts the focus on the point of establishing temporal correspondence among audio and visual modalities. The main methods introduced here is the attention-based module in the Siamese framework and the dynamic layer-wise weighted fusion strategy in the learning process. Experiments on LRS2 and LRS3 show the effectiveness of the proposed method.

**Strengths:**

The general structure is clear. The method is simple in general. It’s easy to follow. The illustration of the motivation and advantages of using each module, e.g. reconstruction masked tokens, the mix-attention encoder, is clear.

**Weaknesses:**

Many involved modules and strategies in this work have been proposed in other works. The encoders, the Siamese framework with target setting not masks and student setting random masks, random masking for different modalities and the optimization functions have all been proposed in previous works. The manner to obtain proposed adaptive multimodal fusion by introducing weights on the visual modality and performing weighted sum over different layers is a little straightforward and common. I am not very clear about the novel contribution of this paper.

**Questions:**

In Table2, the effect of the proposed adaptive multimodal fusion (obtained with a weighted sum over different layers) seems not so appealing, and the slight improvement may come from the increase of extra learnable parameters? Have the authors checked the learned value of the weight S and wi? What are they like in the learning process?

In Table 3, the gap among different strategies can achieve as high as about 2%, which has already been significant if we compared it with the general gap of the final performance of the proposed work in Table1. This big gap, especially when compared with the improvement of the proposed modules here, states the importance of training strategies in the process, but these strategies are existing ones, not firstly proposed in this work. I think this may further weakens the contribution of the proposed method.

---

> ### Author Response · Authors · 2023-11-16
>
> We appreciate your valuable comments. We list the responses to the questions below.
> >In Table2, the effect of the proposed adaptive multimodal fusion (obtained with a weighted sum over different layers) seems not so appealing, and the slight improvement may come from the increase of extra learnable parameters? Have the authors checked the learned value of the weight S and wi? What are they like in the learning process?
>
>  ANS:
> 1. The learned value of the weight S is 0.75. And the w_i is [0.0751, 0.0751, 0.0787, 0.0733, 0.0739, 0.0727, 0.0703, 0.0735, 0.0774,0.0855, 0.1082, 0.1362] for each layer.
> 2. As for the weight value S, we hypothesize that the audio modality plays a dominant role in the AVSR task, while the visual modality can only serve to improve or enhance the audio modality.
> 3. As for the weight values w_i, we observed that in the shallow layers of audio, they primarily represent speaker information, while in the deeper layers of audio, they primarily represent speech content information. The learned values indicate that speech recognition relies more on speech content, but speaker information can also provide some assistance.
> 4. Additionally, to investigate whether the improvement is attributed to the increase in extra learnable parameters, we conducted additional experiments by setting specific values S_i for each layer i. The performance dropped from 4.9 to 5.0 WER.
>
> > In Table 3, the gap among different strategies can achieve as high as about 2%, which has already been significant if we compared it with the general gap of the final performance of the proposed work in Table1. This big gap, especially when compared with the improvement of the proposed modules here, states the importance of training strategies in the process, but these strategies are existing ones, not firstly proposed in this work. I think this may further weakens the contribution of the proposed method.
>
> ANS:
>
> In the newly revised paper version, the number for the ablation table is Table 3. In Table 3, it can be observed that the disparity in performance between Siamese architectures using different training strategies can be as high as 2%. However, it is important to note that these results are obtained under the assumption that other components of the designed model are already optimized for the 'stop gradient' setting. Furthermore, the reason for selecting these four different training settings is the existence of divergent opinions among researchers regarding the most effective approach to train Siamese networks. Different training strategies are suitable for different scenarios.

---

### Author Response · Authors · 2023-11-18

Dear Reviewers, we appreciate your thorough reviews and the insightful feedback provided on our manuscript.
Taking into account the opinions of all four reviewers, we have made revisions to several sections if the paper and uploaded an updated version. Below is a list of an important revision. Additionally, all modified sections are highlighted in blue in the revised  manuscript.

1. Based on the reviewers' feedback regarding the weakness of our paper, we have carefully re-formulated our motivation to further enhance its clarity and precision. Lip movements primarily contribute to the improvement of Automatic Speech Recognition (ASR) in two main scenarios. Firstly, they offer distinct visual cues that aid in differentiating between speech words that sound similar. Secondly, they provide valuable visual cues that enhance speech recognition accuracy in the presence of various types of noise commonly encountered in real-world environments.

2. According to the revised motivation, we added a new experimental table. Table 1 validate the effectiveness of our model in the first scenarios. And we provide the Table 2 in the revised version to validate the effectiveness of our model in the second scenarios.

3. We have addressed the logical and expression errors pointed out by the reviewers.

4. Based on the questions raised by the reviewers, we have added some additional ablation experiments in the appendix.

---

### Meta-Review · Area_Chair_ATCT · 2023-12-07

**Metareview:**

The paper presents a audio-visual speech recognition framework named Siamese Masked Interaction Learning (SMILE) framework, which facilitates global interactions among audio-visual features and enables single-modal and cross-modal local alignment. The design choices are solid and the paper is well written.

The novelty of this work was considered limited and the performance was considered moderate as compared with state of the art.

**Justification For Why Not Higher Score:**

All four confident reviewers consistently rated the paper as "marginally below the acceptance threshold".

**Justification For Why Not Lower Score:**

N/A

---

### Decision · Program_Chairs · 2024-01-16

Reject